

# Acetate turnover and methanogenic pathways in Amazonian lake sediments

Ralf Conrad[a], Melanie Klose[a], Alex Enrich-Prast[b,c]

[a]Max Planck Institute for Terrestrial Microbiology, Karl-von-Frisch-Str. 10, 35043 Marburg, Germany
[b]Department of Thematic Studies - Environmental Change, Linköping University, Linköping, Sweden
[c]Departamento de Botânica, Instituto de Biologia, University Federal do Rio de Janeiro (UFRJ), Rio de Janeiro, Brazil

Corresponding author:
Ralf Conrad, Max Planck Institute for Terrestrial Microbiology, Karl-von-Frisch-Str. 10, 35043 Marburg, Germany
Tel. +49-6421-178801; Fax: +49-6421-178809;
email: Conrad@mpi-marburg.mpg.de





**Abstract**
Lake sediments in Amazonia are a significant source of $CH_4$, a potential greenhouse gas.
Previous studies of sediments using $^{13}C$ analysis found that the contribution of
hydrogenotrophic versus aceticlastic methanogenesis to $CH_4$ production was relatively high.
Here, we determined the methanogenic pathway in the same sediments (n = 6) by applying
$[^{14}C]$bicarbonate or $[2-^{14}C]$acetate, and confirmed the high relative contribution (50-80%) of
hydrogenotrophic methanogenesis. The respiratory index (RI) of $[2-^{14}C]$acetate, which is
$^{14}CO_2$ relative to $^{14}CH_4 + {}^{14}CO_2$, divided the sediments into two categories, i.e., those with an
RI < 0.2 being consistent with the operation of aceticlastic methanogenesis, and those with an
RI > 0.4 showing that a large percentage of the acetate-methyl was oxidized to $CO_2$ rather
than reduced to $CH_4$. Hence, part of the acetate was probably converted to $CO_2$ plus $H_2$ via
syntrophic oxidation, thus enhancing hydrogenotrophic methanogenesis. This happened
despite the presence of potentially aceticlastic *Methanosaetaceae* in all the sediments.
Alternatively, acetate may have been oxidized with a constituent of the sediment organic
matter (humic acid) serving as oxidant. Indeed, apparent acetate turnover rates were larger
than $CH_4$ production rates except in those sediments with a R < 0.2. Our study demonstrates
that $CH_4$ production in Amazonian lake sediments was not simply caused by a combination of
hydrogenotrophic and aceticlastic methanogenesis, but probably involved additional acetate
turnover.



## 1. Introduction


Acetate is an important intermediate in the anoxic degradation of organic matter and is
produced by fermentation processes and chemolithotrophic homoacetogenesis (Conrad, 2019;
Conrad et al., 2014; Ye et al., 2014). The contribution of these two processes to acetate
production is difficult to determine, but seems to be quite different for different environments
(Fu et al., 2018; Hädrich et al., 2012; Heuer et al., 2010). The degradation of acetate requires
a suitable oxidant such as oxygen, nitrate, ferric iron or sulfate. If such oxidants are not or no
longer available, such as in many freshwater environments (e.g., paddy fields, lake sediments,
peat) acetate sometimes accumulates until suitable electron acceptors become again available.
Temporal accumulation and subsequent oxidative consumption has for example been
observed in peatlands during increase and decrease, respectively, of the water table
(Duddleston et al., 2002). Normally, however, acetate degradation in the absence of inorganic
electron acceptors is accomplished by aceticlastic methanogenesis (Conrad, 2019). If
aceticlastic methanogenesis is operative, the methyl group of the acetate is converted to $CH_4$.
If methanogenesis is the exclusive final step in the anaerobic degradation of organic
matter, polysaccharides (one of the most important compounds from primary production) will
be dismutated to equal amounts of $CH_4$ and $CO_2$. Furthermore, acetate usually accounts for
more than two third of total methane production, especially if polysaccharides are the
predominant degradable organic matter (Conrad, 2019). However, $CO_2$ has often been found
to be the main product in many anoxic environments despite the absence of inorganic electron
acceptors ($O_2$, nitrate, ferric iron, sulfate) (Keller et al., 2009; Yavitt and Seidmann-Zager,
2006). Such results have been explained by the assumption that organic substances (e.g.
humic acids) may also serve as electron acceptors (Gao et al., 2019; Keller et al., 2009;
Klüpfel et al., 2014). Organic electron acceptors also allow the oxidation of acetate (Coates et
al., 1998; Lovley et al., 1996). The role of organic electron acceptors during anaerobic
degradation of organic matter is potentially important, but still not well known (Corbett et al.,

47   2013)

There are also many reports that methane production in lake sediments is dominated by
hydrogenotrophic rather than aceticlastic methanogenesis (Conrad et al., 2011; Ji et al., 2016).





Such observations were explained (1) by incomplete degradation of organic matter producing
predominantly $H_2$ and $CO_2$ without concomitant acetate production (Conrad et al., 2010;
Hodgkins et al., 2014; Liu et al., 2017), (2) by acetate oxidation coupled to the reduction of
organic substances (see above), or (3) by syntrophic acetate oxidation coupled with
hydrogenotrophic methanogenesis (Lee and Zinder, 1988; Vavilin et al., 2017). If syntrophic
acetate oxidation is operative, the methyl group of the acetate is converted to $CO_2$, similarly
as found during acetate oxidation with external inorganic or organic electron acceptors.
However, syntrophic acetate oxidation does not require a chemical compound (other than $H^+$)
as electron acceptors, since it is the hydrogenotrophic methanogenesis that eventually accepts
the electrons released during acetate oxidation.
Syntrophic acetate oxidation can replace aceticlastic methanogenesis and thus, has been
found when aceticlastic methanogenic archaea were not present in the microbial community
of lake sediment (Nüsslein et al., 2001). This may also happen in other anoxic environments
when conditions are not suitable for aceticlastic methanogens, e.g., at elevated temperatures
(Conrad et al., 2009; Liu and Conrad, 2010; Liu et al., 2018), in the presence of high
concentrations of ammonium (Müller et al., 2016; Schnürer et al., 1999; Zhang et al., 2014),
or phosphate (Conrad et al., 2000). However, syntrophic acetate oxidation has also been found
in lake sediments that contained populations of putatively aceticlastic methanogens (Vavilin
et al., 2017). It is presently unkown under which conditions syntrophic acetate oxidizers can
successfully compete with aceticlastic methanogens and co-occur with acetate oxidation that
is coupled to the reduction of organic substances.
As a further step in understanding the ecology of syntrophic acetate oxidizers versus
aceticlastic methanogens, we attempted to document their coexistence by studying lake
sediments, which had been reported containing 16S rRNA genes of putatively aceticlastic
*Methanosaetaceae* (*Methanotrichaceae* (Oren, 2014)) (Ji et al., 2016). We used these
sediments and measured the fractions of hydrogenotrophic methanogenesis and of the methyl
group of acetate being oxidized to $CO_2$ rather than reduced to $CH_4$, and compared the
turnover of acetate to the production rate of $CH_4$.





## 2. Materials and Methods

The sediment samples were obtained from floodplain lakes in the Amazon region and have
already been used for a study on structure and function of methanogenic microbial
communities (Ji et al., 2016). In particular, these sediment have been assayed for the
percentage of hydrogenotrophic methanogenesis using values of $\delta^{13}C$ of $CH_4$, $CO_2$ and
acetate-methyl, and for the percentage contribution of putatively aceticlastic methanogens to
the total archaeal community (Ji et al., 2016). Here, we used six of these sediments for
incubation experiments with radioactive tracers. These sediments are identical to those listed
in our previous publication (Ji et al., 2016). The identity of the lake sediments and the
percentage content of putatively aceticlastic methanogens is summarized in Table 1.
The experiments were carried out at the same time as those in our previous publication and
were basically using the same incubation techniques. Briefly, for determination of $CH_4$
production rates and the fractions of hydrogenotrophic methanogenesis  about 10-15 ml of
each replicate (n =3) were filled into 27-ml sterile tubes, flushed with $N_2$, closed with butyl
rubber stoppers, and incubated at 25ºC. After preincubation for 12 days (in order to deplete
eventually present inorganic oxidants), 0.5 ml of a solution of $NaH^{14}CO_3$ (about 1 µCi) was
added, the tubes flushed again with $N_2$, and incubation was continued at 25ºC for about 100
days. Partial pressures of $CH_4$ and $CO_2$ as well as their contents of $^{14}C$ were measured at
different time points after mixing the slurries by heavy manual shaking. The gas partial
pressures were measured by gas chromatography with a flame ionization detector (Ji et al.,
2016), the radioactivities were analyzed with a radiodetector (RAGA) (Conrad et al., 1989).
For determination of acetate turnover, the same conditions were used, except that
preincubation was for 25 days, 0.5 ml of a solution of carrier-free $Na[2-^{14}C]$acetate (about 2
µCi) was added, and incubation was continued for about 8 h. During this time gas samples
were repeatedly taken and the radioactivities in $CH_4$ and $CO_2$ were analyzed in a gas
chromatograph with a radiodetector (RAGA) (Conrad et al., 1989). In the end, the sediment
samples were acidified with 1 ml of 1M $H_2SO_4$ to liberate $CO_2$ from carbonates, and the
radioactivities in $CH_4$ and $CO_2$ were analyzed again.



The data were used to calculate the fractions of hydrogenotrophic methanogenesis ($f_{H2}$),
the acetate turnover rate constants ($k_{ac}$) and the respiratory index (RI) values as described by
Schütz et al. ( 1989). The RI is defined as RI = $^{14}CO_2/(^{14}CO_2 + ^{14}CH_4)$. The acetate turnover
rates were calculated as the product of $k_{ac}$ time the acetate concentration, which was analyzed
in the sediments at the end of the incubation using high pressure liquid chromatography. The
acetate concentrations are summarized in Table 1.

**3.     Results**
Six different lake sediments from Amazonia were incubated in the absence and the
presence of $H^{14}CO_3$. Although the rates of $CH_4$ production were different in the two different
incubations, the orders of magnitude were similar for the different lake sediments (Fig. 1A).
The incubations in the presence of $H^{14}CO_3$ were used to follow the specific radioactivities of
$CH_4$ (Fig. 2A) and $CO_2$ (Fig. 2B) over the incubation time. The specific radioactivities of $CH_4$
changed only little but were slightly different for the different lake sediments. The specific
radioactivities of $CO_2$ decreased with time as expected due to the production of non-
radioactive $CO_2$. Both specific radioactivities were used to calculate the fractions of
hydrogenotrophic methanogenesis ($f_{H2}$), which increased with incubation time and eventually
reached a plateau. The values of $f_{H2}$ averaged between 30 and 60 d of incubation are
summarized in Fig. 1B. Only the incubations of sediment "Grande" did not reach a plateau
but still increased after 260 d of incubation due to the continuously decreasing specific
radioactivities of $CO_2$ (data not shown). Averaging these values over the 4 data points
between 160 and 260 d resulted in $f_{H2}$ of about 60% (Fig. 1B). The thus determined values of
$f_{H2}$ were comparable to those determined in the absence of $H^{14}CO_3$ using values of $\delta^{13}C$,
which have already been published (Ji et al. 2016) (Fig.1B).
The same sediments were used to determine the turnover of $[2-^{14}C]$acetate by measuring
the increase of radioactive $CH_4$ (Fig. 3A) and $CO_2$ (Fig. 3B). These data were used to
determine the rate constants of acetate turnover (Fig. 3C), which ranged between 0.02 and 1.7
$h^{-1}$. The respiratory indices (RI) were generally larger than 0.2 except those of the sediments
Tapari and Verde, which were smaller than 0.2 (Fig. 4A). The RI values and the acetate



turnover rate constants were used to calculate the rates of $CH_4$ production from acetate in
comparison to the rates of total $CH_4$ production (Fig. 4B). Interestingly, acetate-dependent
$CH_4$ production was always larger than total $CH_4$ production, except in those sediments
exhibiting a RI <0.2.

**4.        Discussion**

The RI value quantifies the fraction of the methyl group of acetate that is oxidized to $CO_2$

rather than reduced to $CH_4$. Since some oxidation of acetate methyl is also happening in pure
cultures of aceticlastic methanogens (Weimer and Zeikus, 1978), and since a RI of around 0.2
has often been found in environments where acetate turnover was dominated by aceticlastic
methanogenesis (Phelps and Zeikus, 1984; Rothfuss and Conrad, 1993; Winfrey and Zeikus,
1979), an RI value of 0.2 may in practice be used as the threshold for the change of
methanogenic to oxidative acetate turnover. Based on this criterion, i.e. RI < 0.2, the lake
sediments of Tapari and Verde behaved as when acetate turnover was exclusively caused by
aceticlastic methanogenesis. The percentage of acetate-dependent $CH_4$ production was fairly
consistent with the fraction of hydrogenotrophic methanogenesis, which made up the
remainder of total $CH_4$ production. In conclusion, the acetate turnover and $CH_4$ production in
these lake sediments behaved as expected as when aceticlastic methanogenesis was the sole
process of acetate consumption (Fig. 5).

However, the sediments of Jua and in particular those of Jupinda, Cataldo, and Grande

exhibited RI values >0.2, showing that a substantial fraction of the acetate-methyl was
oxidized to $CO_2$. Hence, acetate was not exclusively consumed by aceticlastic
methanogenesis, but it was oxidized, for example by syntrophic acetate oxidation producing
$H_2$ and $CO_2$. The $H_2$ and $CO_2$ may subsequently have been used as methanogenic substrates,
thus supporting $CH_4$ production (Fig. 5). Such support would be consistent with the relatively
high fractions ($f_{H2}$) of hydrogenotrophic methanogenesis observed in these sediments.
However, it would not explain why acetate turnover rates were higher than necessary for
supporting the observed rates of total $CH_4$ production. A possible conclusion is that acetate
was converted to $CO_2$ without concomitant production of $H_2$. Possibly, electrons from acetate



were transferred to organic electron acceptors (Fig. 5), such as suggested in the literature
(Coates et al., 1998; Lovley et al., 1996). In conclusion, these lake sediments behaved as
when acetate consumption was accomplished not only by aceticlastic methanogenesis, but
also by oxidative consumption.

Our conclusions are mainly based on radiotracer measurements, which may be biased. For

example, acetate turnover rate constants are calculated from acetate concentrations and
turnover rate constants. Acetate concentrations were only measured at the end of incubation
and thus, may not have been representative for much of the incubation time. Furthermore,
acetate in the sediment may occur in several pools with different turnover (Christensen and
Blackburn, 1982). Therefore, acetate turnover rates and acetate-dependent $CH_4$ production
rates may be overestimated, if the actual acetate turnover depends on a pool size that is
smaller than that analyzed. Such overestimation in sediments of Jua, Jupinda, Cataldo and
Grande cannot be completely excluded, although rates in sediments of Tapari and Verde were
in a realistic range.

The determination of fractions of hydrogenotrophic methanogenesis ($f_{H2}$) depends on the

specific radioactivity of the dissolved $CO_2$ pool that is involved in $CH_4$ production. However,
it is the pool of gaseous $CO_2$ that is analyzed in the assay, assuming that its specific
radioactivity is identical to that of the active dissolved pool. Since non-radioactive $CO_2$ is
permanently produced from oxidation of organic matter, there may be disequilibrium.
Nevertheless, determinations of $f_{H2}$ using radioactive bicarbonate exhibited the same
tendencies as those based on $\delta^{13}C$ values, and thus are probably quite reliable.

Despite these reservations, our results collectively demonstrated that acetate turnover in

tropical lake sediments did not necessarily follow a canonical pattern with aceticlastic
methanogenesis as sole or predominant process of acetate turnover, despite the fact that all
these sediments contained populations of putative aceticlastic methanogenic archaea. Acetate
consumption in *Methanosaeta* species is known to have a relatively high affinity and a low
threshold for acetate (Jetten et al., 1992). Therefore, the question arises why oxidative
processes, including syntrophic acetate oxidation, could successfully compete with
aceticlastic methanogenesis.




## 5. Author contribution

RC designed the experiments, evaluated the data and wrote the manuscript; MK did the
experiments; AEP provided the samples and contributed to the discussion of the data.

## 6. Competing interests

The authors declare that they have no conflict of interest.

## 7. Acknowledgements

AEP acknowledges funding from the Swedish Research Council Vinnova and Linköping
University and for funding from the Brazilian Research Council FAPERJ.

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





Table 1: Identity of sediment samples (same as those in Ji et al. (2016)),
percentage content of putatively aceticlastic methanogens (*Methanosaetaceae*)
relative to total archaea, and concentrations of acetate; mean ± SE.

| Lake # | Name | Type | *Methanosaetaceae* (%) | Acetate (nmol g⁻¹ dry weight) |
|---|---|---|---|---|
| P1 | Jua | clear water | 21 ± 1 | 93 ± 5 |
| P8 | Tapari | clear water | 19 ± 3 | 261 ± 39 |
| P9 | Verde | clear water | 19 ± 11 | 126 ± 12 |
| P10 | Jupinda | clear water | 27 ± 4 | 110 ± 6 |
| A1 | Cataldo | white water | 42 ± 1 | 50 ± 3 |
| A2 | Grande | white water | 36 ± 3 | 35 ± 1 |




**Figure captions**

Fig. 1: Methane production in sediments of different Amazonian lakes: (A) rates of $CH_4$ production, and (B) fractions of hydrogenotrophic methanogenesis, both determined in the absence and the presence of radioactive bicarbonate. The data in the absence of radioactive bicarbonate are the same as published in Ji et al. (2016), when $f_{H2}$ was determined from values of $\delta^{13}C$; mean $\pm SE$.

Fig. 2: Conversion of radioactive bicarbonate in sediments of different Amazonian lakes: (A) specific radioactivities in $CH_4$; (B) specific radioactivities in gaseous $CO_2$; and (C) fractions ($f_{H2}$) of hydrogenotrophic methanogenesis; mean $\pm SE$.

Fig. 3: Conversion of $[2\text{-}^{14}C]$acetate in sediments of different Amazonian lakes: (A) accumulation of radioactive $CH_4$; (B) accumulation of radioactive gaseous $CO_2$; and (C) acetate turnover rate constants; mean $\pm SE$.

Fig. 4: (A) Rates of total and acetate-derived $CH_4$ production in sediments of different Amazonian lakes and (B) respiratory indices (RI) of the turned over $[2\text{-}^{14}C]$acetate; mean $\pm SE$.

Fig. 5: Scheme of the pathways involved in acetate turnover in sediments of Amazonian lakes; (1) aceticlastic methanogenesis; (2) syntrophic acetate oxidation; (3) hydrogenotrophic methanogenesis; (4) acetate oxidation with organic electron acceptors.



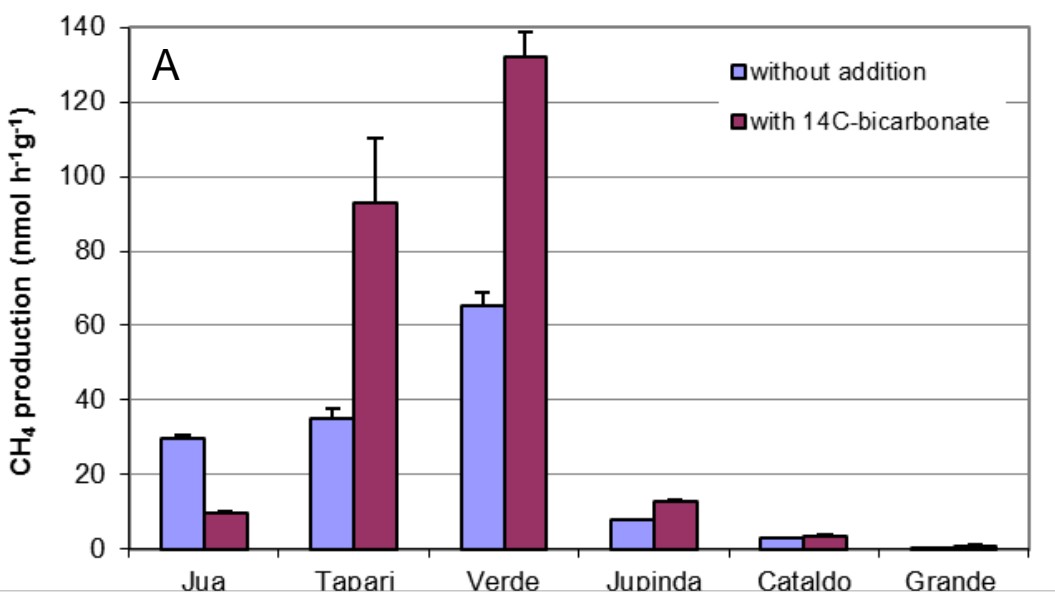

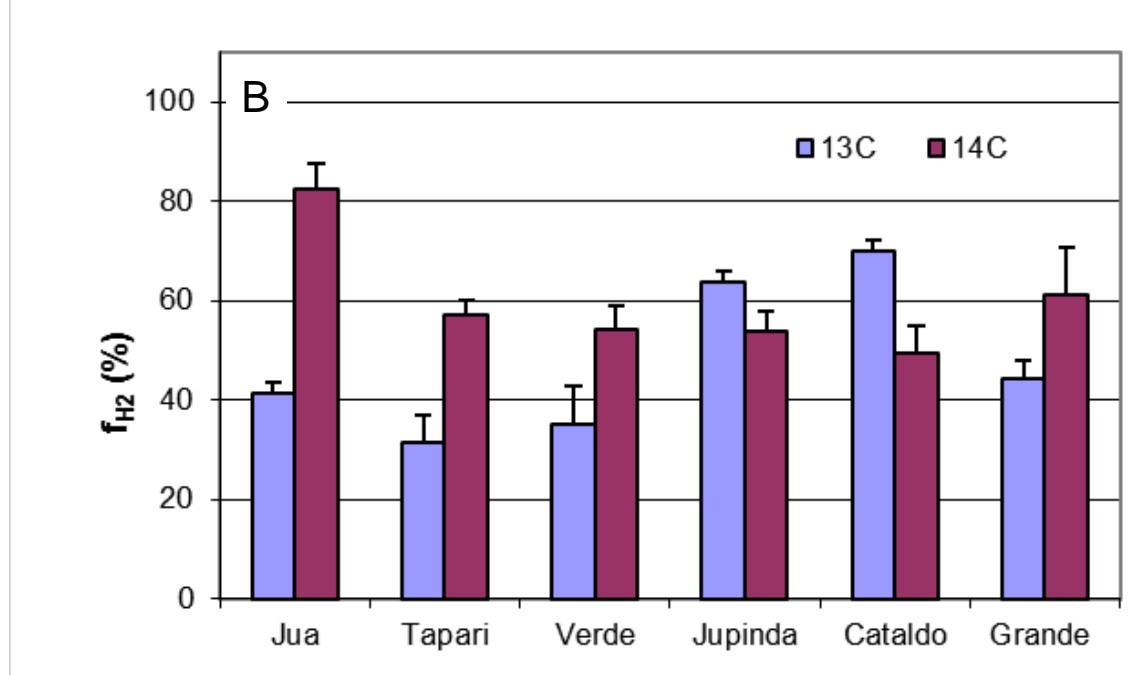

Fig. 1





Fig. 2





Fig. 3



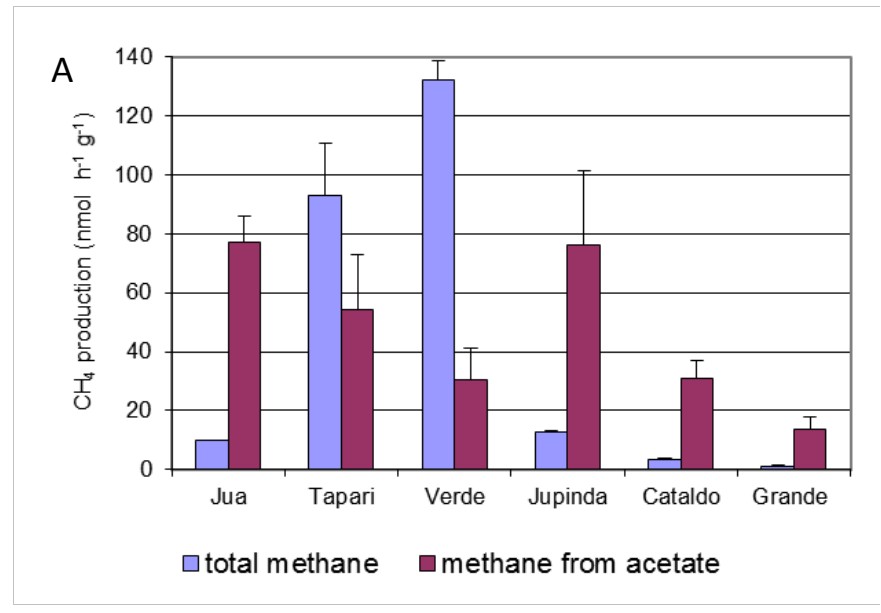

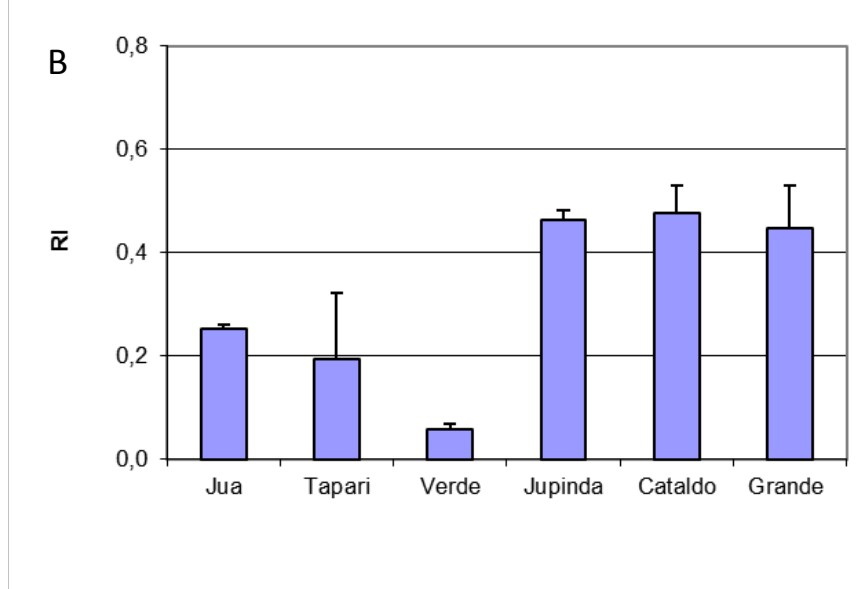

Fig. 4



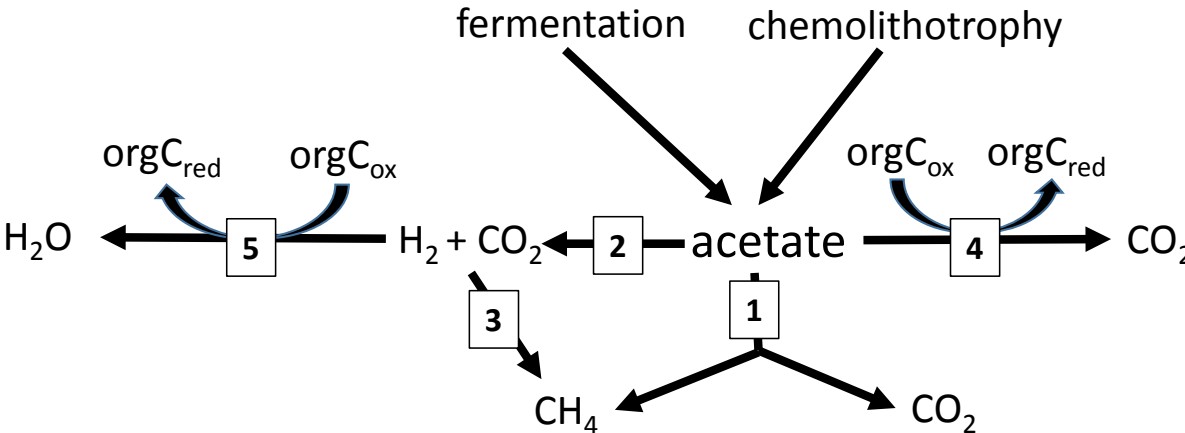

Fig. 5