# Peer review of "Acetate turnover and methanogenic pathways in Amazonian lake sediments"

_Biogeosciences, 2019_

## Referee Comment (RC1) · Anonymous Referee #1 · 31 Oct 2019

The manuscript "Acetate turnover and methanogenic pathways in Amazonian lake sediments" presents data from radioactive C isotope tracer incubations with the aim to identify the pathways responsible for consumption of acetate in freshwater lake sediments under methanogenic conditions. The radiotracer data are compared to results from previously published stable C isotope tracer incubations (Ji et al. 2016) that were performed in parallel to those incubations described in the presented manuscript. The studied lake sediments have been well characterized previously by Ji et al. (2016) and differ in their biogeochemistry, microbial community composition, their rates of CH4 production, as well as their apparent contribution of different methanogenic pathways to overall CH4 production.

In the presented manuscript, Conrad et al. find that an unusually large fraction of

methanogenesis is apparently attributed to CO2-reduction, despite a high acetate turnover and the presence of putative acetoclastic methanogenic microorganisms. Similar to previous studies on methanogenesis in sediments, this is explained by the authors with a potential coupling between acetate oxidation and subsequent consumption of electrons and CO2 by hydrogenotrophic methanogenesis. Additionally, the potential of organic C as a subterminal electron acceptor is briefly discussed. Overall, the paper is short and concise, conclusions are carefully formulated, but findings have little novelty. There are a few concerns that I would like to see adressed in a revised version of the manuscript:

The fraction of methanogenesis from CO2 and/or acetate is estimated via calculation of the parameters "fH2" (calculated either from 14C or 13C incubations), the respiratory index "RI", or the "methane production with 14C-bicarbonate". These parameters differ in their calculation, and the data that goes into these calculations. Even for a reader familiar with the applied methods, the text can be confusing at times, especially with regards to comparability between the methods/incubations. Aggravating the confusion, the manuscript presents several conflicting findings, e.g. higher production of methane from acetate than total methane production or major differences in CH4 production with/without radiotracer, that are not well discussed by the authors. It seems that the authors disregard inconsistencies in their experimental procedures (e.g. different incubation times, different preincubation times) that might very well explain the before-mentioned discrepancies. Maybe the authors assume that the methanogenic communities stabilized over the course of the incubation time. Several studies have shown that this is most likely not the case and even under apparently stable incubation conditions, processes and associated microbial communities can be quite dynamic.

Some minor issues:

General – the authors should think about diversifying their references. About half of the references in the manuscript stem from the author's lab. Some of these are appropriate, others not.

Methods: CH4 production without radiotracer addition is shown in Fig. 1 but not described in the Methods section. How much did the authors change the acetate concentration by the addition of radiotracer?

Fig. 2: Why is only the specific radioactivity over time shown?

L. 32: "Normally" is not appropriate here. I suggest "It is generally assumed that,"

L. 86: Please specify "identical". Same sediment? Same sampling location? Same sampling time? Homogeneous mixture?

L. 106: How much radioactivity was found in the form of carbonates?

L. 124: I don't understand why an average was chosen here.

---

## Author Comment (AC1) · 6 Nov 2019

Reply to anonymous referee #1

We thank the referee for the helpful comments. We are happy to address them in the following and in a revised version of the manuscript:

The referee briefly remarks that the findings have little novelty. This may be the case with the conclusions, which emphasize the potential importance of acetate oxidation coupled either syntrophically to hydrogenotrophic methanogenesis or to the reduction of organic substances. Such conclusions have indeed been made before. However, to our knowledge they have so far not been supported by showing that the methyl group of acetate is indeed oxidized to $CO_2$ rather than reduced to $CH_4$. Here, we addressed

this point by determining the fate of radioactively labeled acetate-methyl, and indeed found that in some of the lake sediments a large percentage of the acetate methyl was oxidized to CO2. In our opinion, this is a crucial and novel experimental result.

The fraction of methanogenesis from either CO2 (hydrogenotrophic) or from acetate (aceticlastic) was determined in three different ways. Two of them determined the fraction of hydrogenotrophic methanogenesis. The first used the fractionation of 13C. This approach had been used in our previous publication (Ji et al. 2016), and the method and the results are described there in detail. This method basically depends on the abundance of 13C in CH4 and in acetate after anaerobic incubation without any amendments or with methyl fluoride, an inhibitor of aceticlastic methanogenesis. The second method was to add 14C-labelled bicarbonate and determine upon incubation the specific radioactivities in the substrate (i.e., CO2) and the product (i.e., CH4) of hydrogenotrophic methanogenesis. This approach was used in the present study and the results were compared to those using the natural abundance of 13C.

However, the measurement of the RI from 14C-labeled acetate-methyl was not done for the determination of the fraction of hydrogenotrophic versus aceticlastic methanogenesis. Instead it served a different purpose, i.e., to determine the fraction of acetate-methyl being oxidized to CO2 or being reduced to CH4. This analysis was important to prove that acetate-methyl had indeed been oxidized. We regret that the text of the methods section confused the purpose of the different determinations. We will revise the text for improvement.

Finally, the rate constants of acetate turnover were determined from the conversion of 14C-labeled acetate-methyl. Together with the RI coefficient, the turnover rate constant allows the calculation of acetate-dependent methanogenesis, which basically is the third way to determine the fractions of methanogenesis from either CO2 or acetate. In the sediments of Tapari and Verde, which have low RI values, the fractions of acetate-dependent CH4 production were consistent with the fH2 values measured by the natural 13C and 14C-bicarbonate methods (see above).

However, in the other sediments with high RI, the determination of acetate turnover resulted in rates of aceticlastic methanogenesis that were actually larger than the rates of total methanogensis. We agree with the referee that this is a conflicting result. The first idea for explanation is of course the assumption of experimental bias. We discussed a few possibilities, but agree that this discussion was probably not exhaustive (we will do better in the revised version). In particular, we should have also emphasized the fact that total rates of CH4 production were measured in different incubations (extending of days) than rates of acetate turnover (extending over hours). The different incubation conditions might give conflicts. However, we still can safely conclude that the fractions of hydrogenotrophic methanogenesis were comparable irrespectively of whether using natural 13C or 14C-bicarbonate, and that in most sediments acetate-methyl was to a large extent oxidized rather than reduced. Hence, the only ambiguity is whether acetate oxidation was coupled to the reduction of organic substrates (processes 4 and 5 in Fig. 5), or was simply the result of syntrophic oxidation coupled to the production of CH4 (processes 2 and 3 in Fig. 5).

Minor issues:

They will all be addressed in a revision.

General: We will work on the reference list as suggested by the referee.

Methods: The CH4 production rates without addition of radiotracers (Fig. 1A) are basically those from our previous study (Ji et al., 2016), analogously to the values of fH2 shown in Fig. 1B.

The addition of 14C was almost carrier free, i.e., the specific radioactivitities of [2-14C]acetate and 14C-bicarbonate were on a range of about 50 Ci/mol. Thus, the addition of 1 $\mu$Ci acetate-methyl was equivalent to an amount of about 20 nmol acetate. Hence, the addition of radioactive acetate was small (<5%) compared to the in-situ concentration of acetate. The same is true for bicarbonate.

Fig. 2: The specific radioactivity is the relevant number for determining fH2. Therefore, we preferred showing these values instead of showing only Bq of CO2 and CH4 without reference to total CH4 and CO2.

L.32: ok

L.86: ok

L.106: There was a significant amount of radioactivity in the sediment carbonates, which has to be considered for the correct determination of the RI. In some of the sediments the released radioactive CO2 more than doubled upon acidification. Actually, we also calculated RI values using the final radioactivity in CO2 before acidification. These RI values were (of course) generally smaller than those after acidification, ranging between RI = 0.05-0.30 compared to RI = 0.06-0.48. The use of the unacidified values of RI for calculation of acetate dependent CH4 production would result in even higher rates than those given in our paper. Hence, the rates of acetate-dependent CH4 production given in our paper were conservative. Nevertheless, the conclusions would be the same irrespectively of using the radioactive CO2 before or after acidification.

L.124: Averages were calculated since the values of fH2 were not constant (Fig. 2C) and we wanted to compare the values with those determined earlier using 13C (comparison in Fig. 1B).

---

## Referee Comment (RC2) · Anonymous Referee #2 · 25 Nov 2019

The manuscript by Conrad and coauthors describes investigations of acetate turnover, and the relative importance of methanogenic pathways (acetoclastic vs hydrogenotrophic methanogenesis) in Amazonian lake sediments. The data present here using radiotracer approach are compared to a parallel study (Ji et al. 2016) conducted with 13C tracer for the incubations of same sediments. The authors found that a large fraction of acetate was oxidized to $CO_2$ rather than reduced to $CH_4$. This was interpreted as the syntrophic oxidation of acetate coupled to hydrogenotrophic methanogenesis. While this study is interesting and relatively novel, I have several comments that need to be addressed:

1) The comparison between different methods, rates analyses, incubations is confusing, because of the lack of detailed information on the calculations, method descriptions

etc.

2) The authors suggested the oxidation of acetate could be attributed to the presence of organic but not inorganic electron acceptors. It seems the original sediments contain ~mM sulfate (>30 $\mu$mol/g at A2) according to Ji et al. 2016. Since acetate can be used by sulfate reducers, I wonder if the authors measured the sulfate concentration after preincubation and if sulfate was completely depleted or not.

3) The turnover of acetate was very fast (Fig. 3C) and RI was much higher than 0.2, suggesting that much of acetate was oxidized to CO2. But this did not necessarily mean the syntrophic oxidation of acetate coupled to hydrogenotrophic methanogenesis. Alternatively, the oxidation of acetate and hydrogenotrophic methanogenesis can be two separate and independent processes. 14CO2 produced from 14C-acetate oxidation could be further converted to 14CH4 via hydrogenotrophic methanogenesis. I think it is possible to estimate the importance of this process based on the turnover of 14C-acetate to 14CO2 and the turnover of 14HCO3- to 14CH4.

Specific comments

Line 4: "aceticlastic" or "acetoclastic"? Both have been used, not sure which one is more often.

Line 107-108: Please specify how fH2 and kac were calculated so that the readers do not need to look for the reference.

Line 135: Should be Fig. 4B. Fig. 4A showed total methane vs methane from acetate.

Please also note the supplement to this comment:
https://www.biogeosciences-discuss.net/bg-2019-411/bg-2019-411-RC2-supplement.pdf

―――――――――――――――――――

**Supplement:**

Review of "Acetate turnover and methanogenic pathways in Amazonian lake sediments ", Conrad et al., 2019.

The manuscript by Conrad and coauthors describes investigations of acetate turnover, and the relative importance of methanogenic pathways (acetoclastic vs hydrogenotrophic methanogenesis) in Amazonian lake sediments. The data present here using radiotracer approach are compared to a parallel study (Ji et al. 2016) conducted with $^{13}$C tracer for the incubations of same sediments. The authors found that a large fraction of acetate was oxidized to $CO_2$ rather than reduced to $CH_4$. This was interpreted as the syntrophic oxidation of acetate coupled to hydrogenotrophic methanogenesis. While this study is interesting and relatively novel, I have several comments that need to be addressed:

1) The comparison between different methods, rates analyses, incubations is confusing, because of the lack of detailed information on the calculations, method descriptions etc.
2) The authors suggested the oxidation of acetate could be attributed to the presence of organic but not inorganic electron acceptors. It seems the original sediments contain ~mM sulfate (>30 µmol/g at A2) according to Ji et al. 2016. Since acetate can be used by sulfate reducers, I wonder if the authors measured the sulfate concentration after preincubation and if sulfate was completely depleted or not.
3) The turnover of acetate was very fast (Fig. 3C) and RI was much higher than 0.2, suggesting that much of acetate was oxidized to $CO_2$. But this did not necessarily mean the syntrophic oxidation of acetate coupled to hydrogenotrophic methanogenesis. Alternatively, the oxidation of acetate and hydrogenotrophic methanogenesis can be two separate and independent processes. $^{14}CO_2$ produced from $^{14}$C-acetate oxidation could be further converted to $^{14}CH_4$ via hydrogenotrophic methanogenesis. I think it is possible to estimate the importance of this process based on the turnover of $^{14}$C-acetate to $^{14}CO_2$ and the turnover of $^{14}HCO_3^-$ to $^{14}CH_4$.

Specific comments

Line 4
"acetoclastic" or "acetoclastic"? Both have been used, not sure which one is more often.
Line 107-108
Please specify how $f_{H2}$ and $k_{ac}$ were calculated so that the readers do not need to look for the reference.
Line 135
Should be Fig. 4B. Fig. 4A showed total methane vs methane from acetate.

---

## Author Comment (AC2) · 5 Dec 2019

We are grateful for the helpful comments of the referee, which we address in the following and in a revised version of the manuscript:

1) The referee criticizes that the comparison between different methods is confusing. The comment is similar to one of referee #1. In the revision we will try to better compare the different methods and rate analyses and better report which incubations gave which results.

2) The referee questions whether sulfate, which was initially present in the sediments, could have contributed to acetate oxidation. Indeed, acetate can also be oxidized with inorganic compounds such as nitrate, ferric iron, sulfate. As long as this happens,

aceticlastic methanogenesis (also hydrogenotrophic methanogenesis) is usually suppressed resulting in a lag phase of CH4 production. To avoid such processes, we preincubated the sediments, so that the CH4 production started without initial lag phase. From previous experiments with various other anoxic environmental samples, in which ferric iron and sulfate were analyzed repeatedly during incubation, we concluded that the inorganic electron acceptors, which were present in the original sediment sample, had all be reduced during the preincubation. This is stated in the manuscript.

3) Acetate oxidation and hydrogenotrophic methanogenesis are indeed two independent processes, that are coupled syntrophically via H2 (or perhaps other electron carriers). The turnover rate of radioactive acetate comprises also such syntrophic acetate oxidation. However, syntrophic acetate oxidation results in stoichiometric amounts of hydrogenotrophically formed CH4. If turnover of radioactive acetate is larger than CH4 production (as was the case for several sediments), the surplus cannot be due to syntrophically produced CH4. Hence this amount of acetate oxidation must be caused by other oxidants, i.e., organic compounds if inorganic ones are not available. The conclusion is summarized in Fig.5, to which we will make better reference in the revision. The oxidation of acetate can either be directly coupled to reduction orgC (reaction 4 in Fig.5) or acetate is oxidized to CO2 plus H2 followed by the oxidation of H2 with orgC (reactions 2 plus 5 in Fig. 5). However, acetate oxidation coupled to hydrogenotrophic methanogenesis (reactions 2 plus 3 in Fig. 5) cannot be larger than CH4 production.

Specific comments: Line 4: we will correct the text using only aceticlastic methanogenesis. Line 107-108: We will add the equations describing how the parameters were calculated. Line 135: Thank you, we will correct

---

## Author Response (AR1)

**Reply to the referees** (see also the respective on-line Discussion for more detail)

**Referee #1**

1. The referee briefly remarks that the findings have little novelty. This may be the case with the conclusions, which emphasize the potential importance of acetate oxidation coupled either syntrophically to hydrogenotrophic methanogenesis or to the reduction of organic substances. Such conclusions have indeed been made before. However, to our knowledge they have so far not been supported by showing that the methyl group of acetate is indeed oxidized to $CO_2$ rather than reduced to $CH_4$. Here, we addressed this point by determining the fate of radioactively labeled acetate-methyl, and indeed found that in some of the lake sediments a large percentage of the acetate methyl was oxidized to $CO_2$. In our opinion, this is a crucial and novel experimental result.

2. The referee criticizes the presentation of the methods, in particular the determination of parameters. We have now presented the relevant methods in more detail including the equations used for calculation. We also better explain the determination of $f_{H2}$ as done in our previous publication (Ji et al. 2016) versus the present one, which used $^{13}C$ versus $^{14}C$ methods, respectively. We have now also added more explicit statements which incubation was used for which determination. This concerns in particular the determination of $f_{H2}$ and rates of $CH_4$ production on the one hand, and of RI and acetate turnover on the other hand. Finally, we have added to the Discussion by arguing about the use of different incubations and different incubation times for the individual parameters.

Minor  issues:

General: We made several changes in the references as suggested by the referee.

Methods: The CH4 production rates without addition of radiotracers (Fig. 1A) are basically those from our previous study (Ji et al., 2016), analogously to the values of $f_{H2}$ shown in Fig. 1B. This has now been explained in the Methods. It is also stated in the figure legend.

The addition of $^{14}C$ was almost carrier free, i.e., the specific radioactivitities of [2-$^{14}C$]acetate and $^{14}C$-bicarbonate were on a range of about 50 Ci/mol. Thus, the addition of 1 μCi acetate-methyl was equivalent to an amount of about 20 nmol acetate. Hence, the addition of radioactive acetate was small (<5%) compared to the in-situ concentration of acetate. The same is true for bicarbonate. This information has been added to the Methods.

Fig. 2: The specific radioactivity is the relevant number for determining fH2. Therefore, we preferred showing these values instead of showing only Bq of $CO_2$ and $CH_4$ without reference to total $CH_4$ and $CO_2$.

L.32: ok

L.86: ok

L.106: There was a significant amount of radioactivity in the sediment carbonates, which has to be considered for the correct determination of the RI. In some of the sediments the released radioactive $CO_2$ more than doubled upon acidification. Actually, we also calculated RI values using the final radioactivity in $CO_2$ before acidification. These RI values were (of course) generally smaller than those after acidification, ranging between RI = 0.05-0.30 compared to RI = 0.06-0.48. The use of the non-acidified values of RI for calculation of acetate dependent $CH_4$ production would result in even higher rates than those given in our paper. Hence, the rates of acetate-dependent $CH_4$ production given in our paper were conservative. A brief statement has been added to the Discussion.

L.124: Averages were calculated since the values of $f_{H2}$ were not constant (Fig. 2C) and we wanted to compare the values with those determined earlier using $^{13}C$ (comparison in Fig. 1B). No changes were made.

**Referee #2**

1. The referee criticizes that the comparison between different methods is confusing. The comment is similar to one of referee #1. See there for action taken.

2. The referee questions whether sulfate, which was initially present in the sediments, could have contributed to acetate oxidation. To avoid sulfate reduction, we preincubated the sediments. This information is given in the Methods. We have now also briefly mentioned it in the Results and have emphasized that $CH_4$ production started without lag phase indicating that suppressive concentrations of sulfate (or Fe(III)) were not available.

3. Acetate oxidation and hydrogenotrophic methanogenesis are indeed two independent processes, that are coupled syntrophically via $H_2$ (or perhaps other electron carriers). The turnover rate of radioactive acetate comprises also such syntrophic acetate oxidation. However, syntrophic acetate oxidation results in stoichiometric amounts of hydrogenotrophically formed $CH_4$. If turnover of radioactive acetate is larger than $CH_4$ production (as was the case for several sediments), the surplus cannot be due to syntrophically produced $CH_4$. Hence this amount of acetate oxidation must be caused by other oxidants, i.e., organic compounds if inorganic ones are not available. The conclusion is summarized in Fig. 5, to which we will make better reference in the revision. The oxidation of acetate can either be directly coupled to reduction orgC (reaction 4 in Fig.5) or acetate is oxidized to $CO_2$ plus $H_2$ followed by the oxidation of H2 with orgC (reactions 2 plus 5 in Fig. 5). However, acetate oxidation coupled to hydrogenotrophic methanogenesis (reactions 2 plus 3 in Fig. 5) cannot be larger than $CH_4$ production. We have added a few sentences to the Discussion for clarity. We have also amended the references to Fig. 5 with the respective reactions shown in the scheme of pathways.

 Specific comments:

Line 4: we have used only the term aceticlastic.

Line 107-108: We have added the equations describing how the parameters were calculated.

Line 135: Thank you, we have corrected the figure numbers.

[revised manuscript text omitted]

---

## Author Response (AR2)

**Associate Editor Decision: Publish subject to technical corrections** (29 Jan 2020) by Tina Treude
Comments to the Author:
Dear Ralf and Co-Workers, the reviewer is very pleased with the revision and suggests acceptance of the manuscript. I agree with this suggestion. However, in a personal statement the reviewer pointed to me that the following relevant citations seem to be missing:

Nüsslein et al. 2001 (Environmental Microbiology 3(7), 460–470)
Beulig et al. 2018 (ISME Journal (2019) 13:250–262)
Beulig et al. 2018 (PNAS vol. 115 | no. 2 | 367–372)

All three studies already documented that methyl-C of the 14C-labelled acetate is oxidized to CO2 rather than reduced to CH4. It would therefore be appropriate to give credit to these studies. I agree and suggest to place the respective citations at appropriate locations either in your introduction or discussion.

Let me know in case you have any questions.
Best
Tina

**Response**

Dear Tina,

Thank you for the hint to the references.

In fact, Nüsslein et al. 2001 has been cited in the Introduction. We did not include the reference to Beulig et al. 2018, since the work is dealing with a seabed sediment rather than a lake sediment. Nevertheless, both references are now also included in the Discussion (see marked ms). However, we did not include the Beulig et al. reference in the ISME journal, since this work is mainly dealing with anaerobic methane oxidation, and the data on acetate oxidation are not thus obvious in this paper.

Best wishes

Ralf

[revised manuscript text omitted]